# Noise modulation in retinoic acid signaling sharpens segmental boundaries of gene expression in the embryonic zebrafish hindbrain

Julian Sosnik[1,2,3], Likun Zheng[2,4], Christopher V Rackauckas[2,4], Michelle Digman[1,2,5], Enrico Gratton[1,2,5], Qing Nie[1,2,4], Thomas F Schilling[1,2]*

[1]Department of Developmental and Cell Biology, University of California, Irvine, Irvine, United States; [2]Center for Complex Biological Systems, University of California, Irvine, Irvine, United States; [3]Department of Interdisciplinary Engineering, Wentworth Institute of Technology, Boston, United States; [4]Department of Mathematics, University of California, Irvine, Irvine, United States; [5]Department of Biomedical Engineering, University of California, Irvine, Irvine, United States

**Abstract** Morphogen gradients induce sharply defined domains of gene expression in a concentration-dependent manner, yet how cells interpret these signals in the face of spatial and temporal noise remains unclear. Using fluorescence lifetime imaging microscopy (FLIM) and phasor analysis to measure endogenous retinoic acid (RA) directly in vivo, we have investigated the amplitude of noise in RA signaling, and how modulation of this noise affects patterning of hindbrain segments (rhombomeres) in the zebrafish embryo. We demonstrate that RA forms a noisy gradient during critical stages of hindbrain patterning and that cells use distinct intracellular binding proteins to attenuate noise in RA levels. Increasing noise disrupts sharpening of rhombomere boundaries and proper patterning of the hindbrain. These findings reveal novel cellular mechanisms of noise regulation, which are likely to play important roles in other aspects of physiology and disease.

*For correspondence: tschilli@uci.edu

**Competing interests:** The authors declare that no competing interests exist.

## Introduction

Cells responding to signals need to be able to distinguish these signals from random fluctuations (i.e., noise) and presumably have evolved mechanisms to do so. Noise is inherent in biological systems, but until recently we have lacked the tools to study such complexity in vivo (*Gregor et al., 2007*; *Holloway et al., 2011*). Noise in signaling pathways arises from many sources, including stochastic variation in transcription, protein synthesis, and cellular environment (*Elowitz et al., 2002*). Morphogens are long-range signals thought to induce different cell behaviors in a concentration-dependent manner, but how such graded signals can be established in the face of noise and how they specify sharp boundaries of target gene expression remain unclear.

Retinoic acid (RA) is thought to act as a morphogen in the embryonic vertebrate hindbrain to pattern cells into a series of segments, called rhombomeres, which give rise to different domains of the adult brainstem (*White and Schilling, 2008*; *Niederreither and Dollé, 2008*). A small hydrophobic signaling molecule derived from dietary precursor Vitamin A, RA is produced in mesoderm near the head/trunk boundary and forms an anteriorly declining gradient across the hindbrain progenitor domain, in part through the activities of RA-degrading enzymes, Cyp26s (*Sirbu et al., 2005*;

**eLife digest** Animal cells need to be able to communicate with each other so that they can work together in tissues and organs. To do so, cells release signaling molecules that can move around within a tissue and be detected by receptors on other cells.

We tend to assume that the signaling molecules are evenly distributed across a tissue and affect all the receiving cells in the same way. However, random variations (noise) that affect how many of these molecules are produced, how they move through the space between cells and how they bind to receptors makes the reality much more complex. Cells responding to the signal somehow can ignore this noise and establish sharp boundaries between different cell types so that neighboring cells have distinct roles in the tissue. Few studies have attempted to measure such noise or address how cells manage to respond to noisy signals in a consistent manner.

Retinoic acid is a signaling molecule that plays an important role in the development of the brain in animal embryos. It forms a gradient along the body of the embryo from the head end to the tail end, but it has proved difficult to measure this gradient directly. Sosnik et al. exploited the fact that this molecule is weakly fluorescent and used microscopy to directly detect it in zebrafish embryos. The experiments show that retinoic acid forms a gradient in the embryos, with high levels at the tail end and lower levels at the head end.

Sosnik et al. also found that there is a large amount of noise in the retinoic acid gradient. Two cells in the same position can have very different retinoic acid levels, and the levels in a particular cell can vary from one minute to the next. The experiments also show that proteins that interact with retinoic acid help to reduce noise within a cell. This noise reduction is important for sharpening the boundaries between different brain regions in the embryo to allow the brain to develop normally. A future challenge will be to see if similar retinoic acid gradients and noise control occur in other tissues, and if the noise has any positive role to play in development.

*Hernandez et al., 2007*; *White et al., 2007*). In particular, self-enhanced degradation through induction of Cyp26a1 by RA itself was shown to be critical for gradient formation (*White et al., 2007*). Excess RA during human embryogenesis can cause anterior-posterior (A-P) patterning defects, and RA has been implicated in the development and maintenance of numerous cell types as well as in cancers (leukemia), stem cells (pancreas) and regenerating organs (cardiomyocytes) (*Duester, 2008*; *Tang and Gudas, 2011*; *Rhinn and Dolle, 2012*). Due to its hydrophobic nature, RA requires proteins to bind and transport it through extracellular and intracellular space, which enhances robustness (*Cai et al., 2012*) but also introduces various sources of noise (*Schilling et al., 2012*).

Our computational models suggest that noise in RA signaling can also play a positive role in hindbrain segmentation through noise-induced switches in gene expression at rhombomere boundaries (*Zhang et al., 2012*), but until recently methods were lacking to test this hypothesis. Here we present a novel methodology utilizing fluorescence lifetime imaging microscopy (FLIM) and a phasor analysis (phasor-FLIM) to study the abundance of endogenous RA in vivo in zebrafish embryos. Using this new tool to visualize endogenous RA in living cells we quantify variability in RA levels and provide some of the first evidence that cells actively control the magnitude of noise in a signaling molecule in a multicellular system in vivo.

## Results and discussion

### Fluorescence lifetime imaging (FLIM) measures endogenous RA directly in vivo

We took advantage of the fact that RA is a fluorescent molecule to quantify its endogenous abundance in vivo in the developing zebrafish hindbrain. Due to the low abundance of RA in cells and its wide spectra of absorbance and emission, traditional fluorescence microscopic techniques fail to detect RA specifically. We opted instead to visualize RA by its unique fluorescence lifetime, rather than its fluorescence intensity. Focusing on the presumptive neural ectoderm of mid-gastrula stage embryos (8–

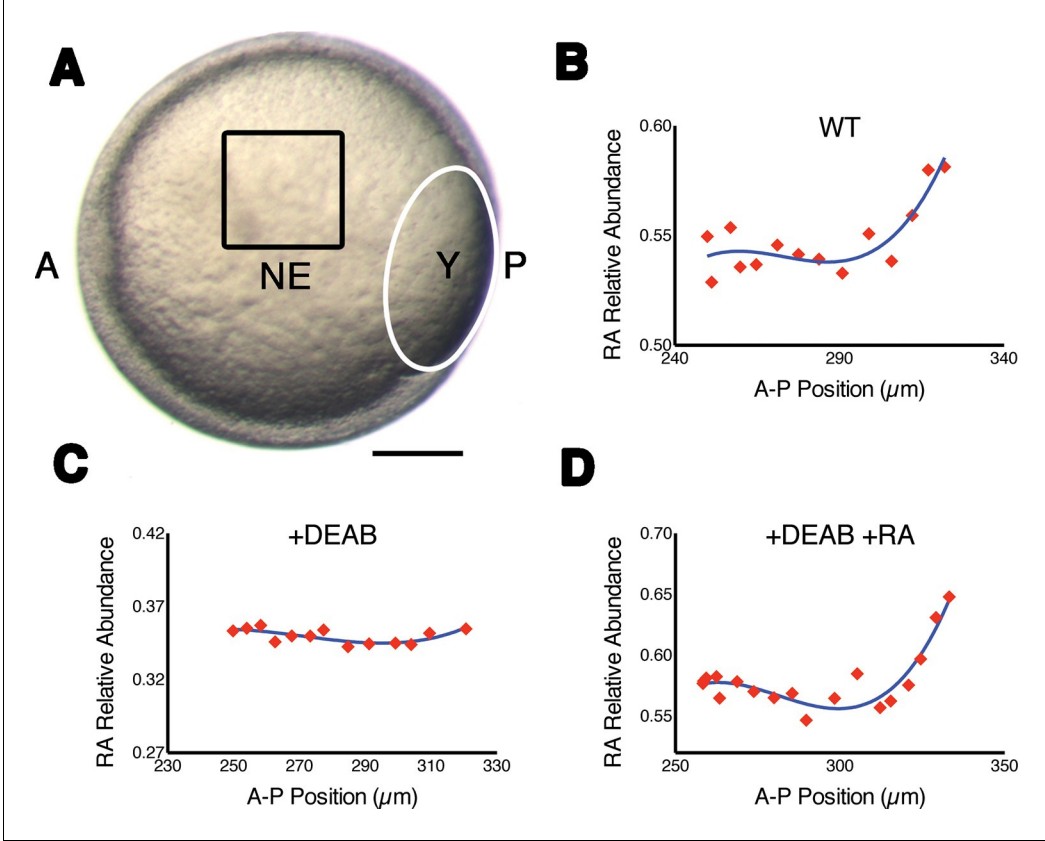

**Figure 1.** Measuring RA gradients in zebrafish embryos with Phasor-FLIM. (**A**) Example of a zebrafish embryo at mid-gastrula stage (8.5 hr post-fertilization) with the imaging area (black square – encompassing positions 230–330 in **B-D**) in the neural ectoderm (NE) centered ~ 200 µm from the advancing blastoderm margin (white line) (A: anterior, P: posterior, Y: yolk). Scale bar = 150 µm. (**B-D**) Plots of the relative abundance of RA (as the difference 1-dRA) versus position in µm along the A-P axis (anterior to the left) in WT (**B**), DEAB-treated (**C**), and DEAB-treated embryos co-treated with 0.7 nM exogenous RA (**D**). Solid curves in (**B-D**) represent best fit.

The following figure supplements are available for figure 1:

**Figure supplement 1.** Phasor-FLIM detects relative levels of RA.

**Figure supplement 2.** Regression analysis of RA gradient shape.

8.5 hr post fertilization) (**Kimmel et al., 1995**) we used FLIM to measure the relative abundances of RA as a function of cell position along the A-P axis. This revealed that intracellular free RA forms an anteriorly-declining gradient (**Figure 1A,B**), similar to that previously reported with FRET reporters for RA (**Shimozono et al., 2013**) and suggested by the pattern of RARE-lacZ expression in late-gastrula mouse embryos (**Sirbu et al., 2005**). Relative abundances were calculated using phasor-FLIM (**Digman et al., 2008**), where within the phasor space, each individual fluorescent species is represented in a characteristic and invariable position. Mixtures of molecules generate a FLIM signature that lies along a line connecting the positions of the individual component species and the position in that line is weighted according to the relative abundances (**Figure 1—figure supplement 1A**). Because we know the absolute position of pure RA (in the lower right corner of phasor space) (**Figure 1—figure supplement 1**)(**Stringari et al., 2011**), we can use the Cartesian distance within this space as a measure of the relative abundance of RA expressed as 1-$d_{RA}$ (**Figure 1—figure supplement 1B**). We observed that 1-$d_{RA}$ increased progressively as measurements were taken further posteriorly within the hindbrain field, suggesting that our FLIM approach is sensitive enough to detect endogenous RA gradients. An ordinary least squares regression analysis of gradient shape could not

distinguish between exponential, linear, and quadratic fits, but confirmed the presence of a gradient (*Figure 1—figure supplement 2*). Unfortunately FLIM is also sensitive to the fluorescence emitted by transgenic markers of rhombomeres or other landmarks in embryos, making it difficult to determine precise segmental locations within the hindbrain field.

To confirm that with phasor-FLIM we could detect RA specifically and its relative levels, we treated embryos with 10 µM DEAB to prevent the enzymatic conversion of retinal to RA, which eliminated the gradient at mid-gastrula stage (*Figure 1C*), and incubating these in 0.7 nM RA re-established the gradient as expected (*Figure 1D*) (*White et al., 2007*). We generated artificial gradients of RA by injecting embryos with RA saturated mineral oil and found that the relative abundance of RA decreased as a function of the distance from the source in two orthogonal axes (*Figure 1—figure supplement 1C*). Phasor analysis of the third harmonic of the laser pulse frequency (240 MHz instead of the standard 80 MHz) revealed similar gradients. Analyzing a different harmonic de-couples $d_{RA}$ from any other component of the mixture – i.e. the locations of each component in the phasor plot will vary independently of changes in $d_{RA}$ - and thus provides an independent means of confirming the specificity of the RA phasor-FLIM signature (*Figure 1—figure supplement 1C*).

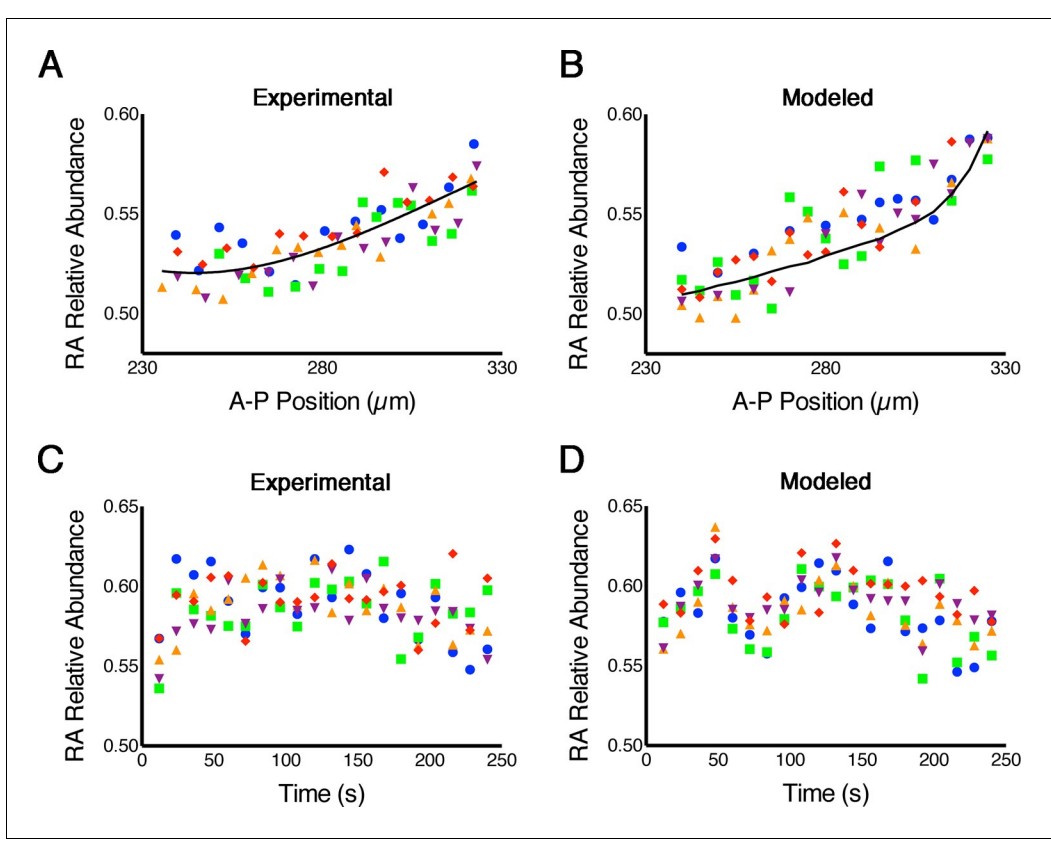

**Figure 2.** RA gradients are noisy in space and time. (**A, B**) Spatial noise. Plots show relative abundance of RA in five parallel rows of cells (each color corresponds to a different row) along the A-P axis of the neural ectoderm within a single embryo. (**A**) Experimental – each point represents the integrated signal of 40 consecutive FLIM measurements (2.7 min) (solid line represents best fit). (**B**) Computational – line represents the mean of 500 model simulations. (**C, D**) Temporal noise. Graphs show variability in relative abundance of RA in five single cells (each color corresponds to a different cell) at equivalent A-P positions over time. (**C**) Experimental – FLIM measurements were taken every 12 s. (**D**) Computational – colors correspond to individual cells for each stochastic realization. See also *Supplementary file 1*.

The following figure supplement is available for figure 2:

**Figure supplement 1.** Instrument noise cannot account for noise in phasor-FLIM measurements of RA.

We next asked if we could detect RA gradients at later stages, when rhombomere boundaries are being established, by performing FLIM measurements in the transgenic line MÜ4127 (Egr2b: mCherry), which labels rhombomeres 3 and 5 (r3, r5), in regions devoid of transgene fluorescence to avoid interference (*Distel et al., 2009*). We found a similar graded increase in $d_{RA}$ on the phasor plot in r4 and r2 relative to r6 at 24 hpf (*Figure 1—figure supplement 1D–F*) (i.e. 1- increases posteriorly), suggesting a graded reduction in RA content anteriorly. Injection of embryos with morpholino oligonucleotides (MOs) targeting Aldh1a2, to inhibit RA synthesis greatly reduced the separation between FLIM signatures in r2, r4 and r6, which was partially rescued by transplantation of wildtype (WT) paraxial mesoderm to restore the local RA source (*Figure 1—figure supplement 1D–F*) (*White et al., 2007*). These results show that the RA gradient persists during gastrulation and establishment of rhombomeres.

## FLIM measurements reveal high noise levels in RA

Because phasor-FLIM measures endogenous RA and is not biased by the $K_d$ of a reporter, in contrast to the FRET method previously published (*Shimozono et al., 2013*), it is more direct and more reliably reflects real-time RA dynamics. Thus we next applied this technique to measure stochastic fluctuations (noise) in RA levels across the embryonic hindbrain. Our models predict that the magnitude of such noise is large (*Lander, 2013*), as we have argued that these fluctuations help cells switch between stable states of gene expression and thereby sharpen gene expression boundaries, i.e. noise-induced switching (*Schilling et al., 2012*; *Zhang et al., 2012*), but direct evidence of such noise is lacking. To assess 'spatial noise' we analyzed five consecutive parallel rows of cells in which each cell within a row lies at the same A-P position within the hindbrain field. This revealed variability as high as 45% of the entire magnitude of the gradient among cells within a row (*Figure 2A*), consistent with the levels of noise predicted by our stochastic mathematical models (*White et al., 2007*; *Zhang et al., 2012*)

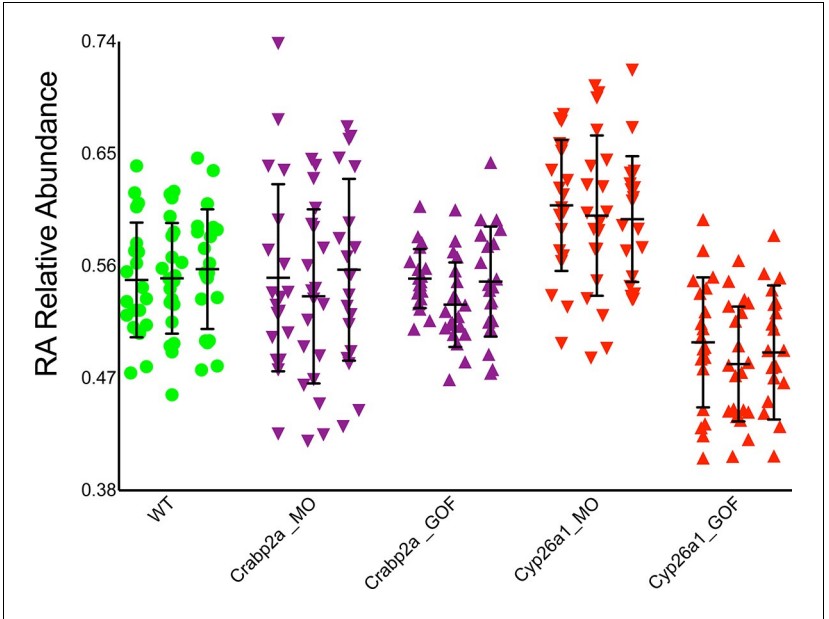

**Figure 3.** Crabp2a but not Cyp26a1 attenuates level of noise in RA. Analysis of the temporal distribution of RA's relative abundance in wildtype (WT), Crabp2a morpholino (MO)-injected, Crabp2a mRNA-injected (gain-of-function - GOF) and Cyp26a1 MO and mRNA-injected zebrafish embryos. Each column shows the signal obtained for a single representative cell and each point corresponds to a single time point. Lines represent the mean and standard deviation. Embryos with reduced or increased levels of Crabp2a show increased and decreased variability in RA, respectively, while altering Cyp26a1 changes the mean concentrations but not the variance.

The following figure supplement is available for figure 3:

**Figure supplement 1.** Crabp2a actively modulates RA signal noise.

(*Figure 2B*; *Supplementary file 1*). To assess 'temporal noise' we analyzed the same cells repeatedly at 12-second intervals, and also found that their RA levels were very noisy (*Figure 2C,D*).

In order to rule out the possibility that the noise in our measurements was introduced by systematic artifacts or the measurement itself, we compared the variance in FLIM measurements of pure solutions of fluorescein, rhodamine and RA with the noise measured in cells of 9 independent embryos and found that noise in embryos is two orders of magnitude greater (*Figure 2—figure supplement 1*). We also calculated the maximum theoretical uncertainty due to photon shot noise and verified that the noise we measured is significantly larger (*Colyer et al., 2008*). Thus the fluctuations in RA levels that we observed in embryos are clearly biological in origin.

Noise in RA levels could be largely irrelevant for downstream gene expression if its frequency is faster than cellular responses, and is therefore averaged out. To address this possibility we performed an autocorrelation analysis of our temporal noise measurements using a moving window on each cell to search for significant lags. This revealed significant correlations (lags 13 and 14) corresponding to a predominant frequency on the order of 2.7 min. This is significantly slower than the half-life of the RA-Crabp2a complex, which is approximately 1.7 min (*Dong et al., 1999*). Because Crabp2 helps deliver RA to its nuclear receptor, and considering the scale of noise in transcriptional activation, noise at this time scale in RA signaling could propagate downstream. Thus it seems likely that cells possess mechanisms to limit this noise propagation.

## Cellular RA binding proteins actively modulate noise

If cells actively control noise in RA signaling, they likely do it through intracellular RA-binding proteins, Crabps, or RA-degrading enzymes, Cyp26s, that can rapidly alter freely available RA (*Kleywegt et al., 1994*) and both of which have been shown to play critical roles in RA signaling (*Sirbu et al., 2005*; *Hernandez et al., 2007*; *White et al., 2007*; *Cai et al., 2012*). To test these candidates we reduced the amount (microinjected MOs) or overexpressed (microinjected mRNA) Crabp2a and Cyp26a1 in zebrafish embryos and measured noise in RA at mid-gastrula stages. Strikingly, MO depletion of Crabp2a increased temporal noise in RA without altering the mean RA level at a given A-P position, while overexpression of Crabp2a decreased variability in RA, again without altering the mean levels of RA (*Figure 3*). In contrast depletion or overexpression of Cyp26a1 increased or decreased mean RA levels, respectively, without altering noise. These results agree with simulations using our stochastic mathematical model in which we altered the levels of Crabp2a or Cyp26a1 (*Figure 3—figure supplement 1*). These results reveal a novel, active role for Crabps in modulating noise in RA.

## Functional roles for noise in RA target gene expression

To determine how altering RA levels influences noise in gene expression within the hindbrain we disrupted Crabp2a or Cyp26a1 and assayed expression of krox20 in r3 and r5. In situ Hybridization Chain Reaction (HCR) (*Choi et al., 2014*) allowed us to quantify krox20 expression levels by manually segmenting confocal images and measuring total fluorescence of each cell (*Figure 4A*) (*Video 1*). All HCR analyses were performed on raw 3D data, with Z-projections performed post-analysis. Either raising or lowering Crabp2a levels increased variance in krox20 expression from cell to cell (*Figure 4B,C*) when normalized for heterogeneity in gene expression from embryo to embryo, as confirmed by single embryo qPCR (*Figure 4—figure supplement 1A*). In addition, it decreased the sharpness of boundaries of krox20 expression in r3 and r5, using a sharpness index calculated as the ratio between the length of the theoretical sharp boundary and the actual measured length of the boundary (*Figure 4D,E*) (*Figure 4—figure supplement 1B*) (Materials and methods). These results suggest that, in contrast to its effects on RA levels where Crabp2a appears to attenuate noise, an optimal range of Crabp2a is required to induce sharp boundaries of gene expression in rhombomeres and too much Crabp2a is also detrimental to the system (*White et al., 2007*; *Zhang et al., 2012*; *Cai et al., 2012*). Similarly, either raising or lowering Cyp26a1 levels increased variance in krox20 expression (*Figure 4*). Thus, while Crabp2a may play a unique role in reducing noise in RA levels it appears to function together with Cyp26a1 and potentially other RA signaling components in allowing robust expression of downstream targets.

Most studies of morphogen gradients and transcriptional noise have focused on the *bicoid-hunchback* transcriptional network in the Drosophila embryo, prior to cellularization and onset of

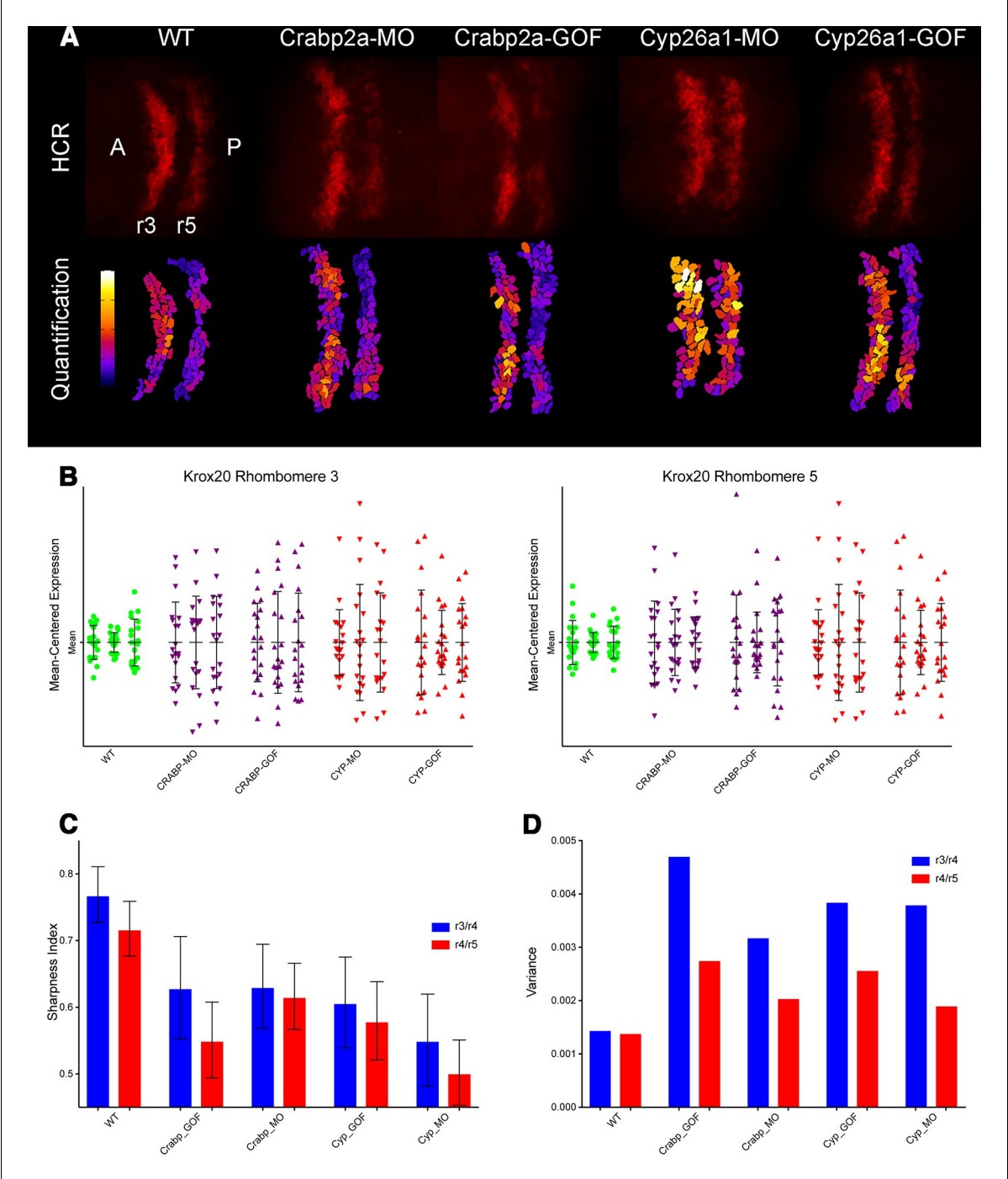

**Figure 4.** Both Crabp2a and Cyp26a1 attenuate noise in *krox20* expression and facilitate rhombomere boundary sharpening. (A) Representative Z projections of r3 and r5 (dorsal views, anterior to the left) analyzed by hybridization chain reaction (HCR) for *krox20* (r3, rhombomere 3; r5, rhombomere 5; A, anterior; P, posterior). We performed all HCR analyses on raw 3D data and later generated Z-projections and enhanced contrast to simplify presentation. Colors correspond to total *krox20* RNA in each cell as measured by total fluorescence intensity bracketed for maximum and minimum for the 5 conditions and represented in a linear scale. (B) Mean-centered analysis of *krox20* expression of a subset of cells for r3 and r5 from 3 randomly selected embryos for each condition. (C) Sharpness indices of the r3/r4 boundary (blue) and r4/r5 boundary (red) for embryos from each of the treatment conditions. Bars correspond to s.d. (D) Analysis of the variance in boundary sharpness from the quantification in (C). All perturbations yielded significant differences from wild-type controls, as noted in the Statistical Analysis. Therefore no asterisks were included to indicate columns representing statistical significance.

The following figure supplement is available for figure 4:

*Figure 4 continued on next page*

*Figure 4 continued*

**Figure supplement 1.** Single embryos show highly variable mean levels of *krox20* expression and boundary sharpness.

zygotic transcription (*He et al., 2012*). The findings in that system indicate that due to the slow diffusion rate of the Bicoid protein, *hunchback* expression is mostly influenced by its own intrinsic noise and transcriptional noise in the *bicoid* gene does not propagate (*Gregor et al., 2007*; *Okabe-Oho et al., 2009*; *Holloway et al., 2011*). In contrast, we show that noise in a secreted signal in the multicellular context of the vertebrate hindbrain influences noise in expression of its transcriptional targets. Our FLIM measurements demonstrate noisy concentration gradients of RA along the A-P axis and reveal a novel role for Crabp2a in noise-attenuation distinct from that of Cyp26a1. Crabp2a could control noise in RA levels rapidly by binding RA and facilitating its entry into cells or buffering its availability within the cytoplasm (*Maden et al., 1989*; *Boylan and Gudas, 1992*) and our previous studies have demonstrated its critical roles in signal robustness (*Cai et al., 2012*). In contrast, both Crabp2a and Cyp26a1 inhibit noise in downstream targets of RA. Previous studies have shown that transcriptional inhibitors act as noise filters within narrow levels of expression, since outside of this range, transcriptional noise in their target genes increases (*Dublanche et al., 2006*). Such a biphasic response resembles our results with Crabp2a and Cyp26a1. Retinoic acid receptors (RARs) often act as transcriptional repressors until they bind RA. Thus Crabp2a and Cyp26a1 may modulate noise in RA targets by altering this balance between activation and repression. As such, both must be present within a narrow optimal range (*Dublanche et al., 2006*; *White and Schilling, 2008*). These mechanisms are likely to be similar in other signaling systems and critical for embryonic development and adult physiology, as well as defective in human diseases.

## Materials and methods

### Reagents

Unless otherwise noted, all of the reagents were obtained from Sigma-Aldrich (St. Louis, MO). All-Trans Retinoic Acid and 4-Diethylaminobenzaldehyde were dissolved at 10 mM and 100 mM, respectively, in anhydrous DMSO to create stocks and kept at -20°C in the dark until used. Morpholino Oligonucleotides (MOs) against *aldh1a2*, *Crabp2a and Cyp26a1* were obtained from Gene Tools (Philomath, OR) and used as previously described (*White et al., 2007*; *Cai et al., 2012*). HCR reagents were obtained from Molecular Tools (Pasadena, CA). Restriction enzymes and SuperScriptII reverse transcriptase kit was obtained from NEB (Ipswich, MA). Light-Cycler 480 SYBR Green I Master mix was obtained from Roche (Indianapolis. IN). mMESSAGE mMACHINE kit, DAPI, Trizol reagent and fluorescein reference standard were obtained from Life Technologies (Eugene, OR).

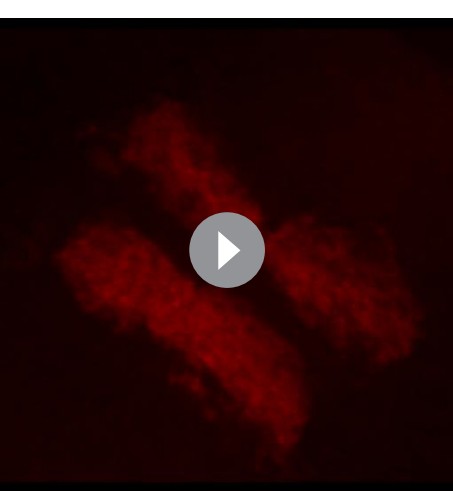

**Video 1.** 3D rendering of HCR dataset. 3D rendering shows the specific HCR signal on rhombomeres 3 and 5 (red) with very low non-specific signal in surrounding tissue, which appears evenly distributed. DAPI signal (blue) demarcates nuclei of cells that are either Krox20 positive (red) or negative (no signal).

### Animals

All animal work was performed under the guidelines of UCI's IACUC. Embryos were obtained by natural crosses, raised in embryo medium (EM), and staged according to *Kimmel et al. 1995*. The AB strain was used for WT experiments. MU4127 transgenics (Tg[shhb:KalTA4,UAS-E1b:mCherry]) to visualize rhombomeres 3 and 5 were kindly provided by Dr. Köster (Helmholtz-Zentrum, München).

## Constructs

For synthesis of mRNA three constructs were generated as templates. pCS2+GFP-CAAX was generated by isothermal assembly of a pCS2+ backbone digested with EcoRI and an amplimer of GFP-CAAX generated by PCR with the primers (forward) 5'-ggatcccatcgattcgTGGACCATGGTGAG-CAAG-3' and (reverse) 5'-gctcgagaggccttgTCAGGAGAGCACACACTTG-3'. Two C terminal Myc-tagged constructs were generated by traditional restriction-ligation procedure using a pCS2+MT as the backbone. Crabp2a was inserted between the BamHI and the ClaI sites of the proximal MCS and Cyp26a1 was inserted between the BamHI and ClaI sites of the proximal MCS. mRNA was synthesized by digestion of the constructs with NotI-HF and in vitro transcription with mMESSAGE mMACHINE SP6 transcription kit.

## Phasor-FLIM

Phasor-FLIM refers to a combination of fluorescence lifetime imaging microscopy (FLIM) and a methodology to analyze FLIM data. Rather than using the traditional intensity of fluorescence to analyze microscopic samples, FLIM measures the lifetime fluorescence decay of the fluorophore. Fluorophores possess a characteristic lifetime of fluorescence that represents the time that takes an excited electron to relax back to its basal state emitting a photon. This technique eliminates most sources of noise present in intensity-based fluorescence microscopy techniques. This is due to the fact that most sources of noise, like thermal flickering or dark current have no lifetimes (*Colyer et al., 2008*). However photon shot noise remains a source of uncertainty, but this is inversely proportional to the square-root of the fluorescence signal intensity. FLIM also requires a high numerical aperture objective (40X/NA 1.2), which intrinsically has a short working distance, making it impossible to perform the measurements at lower magnification.

Representing FLIM data in a phasor plot instead of a time-delay histogram allows analysis of the entire image, rather than pixel by pixel. In addition, because each molecular species is represented in a defined area of the phasor space, it allows analysis of samples with multiple fluorescent species (*Digman et al., 2008*). Individual fluorescent molecules have a constant lifetime, independent of concentration. Because the phasor space operates linearly, analysis of relative concentrations of fluorophores in samples with complex mixes can be performed. Samples with complex mixtures of fluorescent molecules generate a FLIM signature in the phasor plot that corresponds to the linear combination of the positions of the individual fluorescent species in a weighted manner. By calculating the Cartesian distance in the phasor space one can calculate the relative contributions of the different constituents (*Digman et al., 2008*; *Stringari et al., 2011*).

An additional advantage of this method over the use of reporters is the direct measurement of the endogenous fluorophore of interest in vivo and without the potential artifacts introduced by genetic manipulations/transgenic reporters. Genetically encoded FRET reporters published previously for RA, bind RA proportionally to their association/dissociation constants ($k_a/k_d$) and either emit or stop emitting a signal. This binding biases the data (both spatially and temporally) according to the binding constant of the reporter.

## FLIM imaging

Embryos were dechorionated and mounted dorsally on #1.5 coverslips with 1% low-melt agarose in EM without methylene blue. Acquisition and analysis was performed as previously described (*Stringari et al., 2011*). Briefly, the embryos were imaged for 2.7 min (for spatial analysis) or in single frames (4 sec –for temporal analysis) on a Zeiss 710 confocal microscope with a Ti:Sapphire laser (Spectra-Physics, Newport Beach, CA) as a two photon excitation source and an ISS A320 FastFLIM box coupled to two H7422P-40 photo-multiplying tubes (Hamamatsu, Japan). Data acquisition and analysis were performed using SimFCS software (LFD, Irvine, CA). Images were acquired with a 40X 1.2 NA water immersion objective. The excitation frequency used was 760 nm and in order to enrich the signal for RA, the emission was filtered through a 495LP dichroic mirror. Solutions of Rhodamine in water and Fluorescein in 100 mM KOH (pH 9.0) were used as references.

## Embryo manipulations

MOs were injected at the one-cell stage and cell transplantations were performed as previously described (*White et al., 2007*; *Cai et al., 2012*).

For the generation of ectopic retinoic acid (RA) sources, mineral oil was infused with all-trans RA to saturation. Embryos were dechorionated and temporarily mounted in 1% low-melt soft agar in EM over coverslips. Drops of the RA saturated oil or oil alone were then injected in 6–12 embryos using a mouth pipette and a capillary needle. The embryos were then released and left to heal for two hours when they were mounted for FLIM imaging. This experiment was repeated 3 (three) times.

mRNA was injected into one-cell embryos with glass micropipettes and a Narishige IM 300 micro-injector with 50 pg of GFP-CAAX, 50 pg of Crabp2a-Myc or 100 pg of Cyp26a1-Myc. Expression verification was performed by microscopic observation for GFP or by Western blot with anti-Myc antibody (clone 9E10) for Crabp2a-Myc and Cyp26a1-Myc.

## Hybridization chain reaction (HCR)

One-cell stage embryos were injected with mRNA coding for GFP-CAAX to assist in later segmentation. The embryos were then divided into five experimental groups and injected with Crabp2a morpholinos (MOs), Crabp2a-Myc mRNA, Cyp26a1 MOs, Cyp26a1-Myc mRNA or 500 pl of water (WT). Embryos were incubated at 28C in EM until 11 hr postfertilization. Embryos were then treated as previously described (*Choi et al., 2014*). Briefly, embryos were dechorionated and fixed with fresh 4% PFA at 4°C for 16 hr, washed in PBS and dehydrated with methanol for 1 hr followed by graded rehydration. Embryos were then pre-hybridized at 45°C for 30 min in hybridization buffer. Embryos were hybridized in hybridization buffer containing 1 pmol of each of 5 (five) different DNA probes designed against Krox20 containing the B1 double initiator arms and Hoxb1a containing the B2 double initiator arms (*Table 1*) at 45°C for 16 hr. Probe specificity was verified by blast search and controlled by adding the hairpins but no initiator probe, which showed no non-specific signal, as well as single probe experiments. Excess probe was removed and embryos were gradually buffer exchanged to 5xSSCT and washed in 5xSSCT for 3.75 hr at 45°C. Samples were then pre-amplified in amplification buffer for 30 min at room temperature (RT) after which they were left to amplify in amplification buffer containing B1H1 and B1H2 snap-cooled hairpins conjugated to Alexa 594 and B2H1 and B2H2 snap-cooled hairpins conjugated to Alexa 647 at room temperature for 16 hr. Finally the embryos were washed in 5xSSCT and counterstained with DAPI before mounting in soft agar on number 1.5 thickness coverslips for confocal imaging. Samples were imaged with a Leica SP8 scanning confocal microscope acquiring z-stacks covering the entire hindbrain as 12 bit 512 X 512 images and analyzed using ImageJ software. Experiments were performed with 12 embryos per condition and repeated 4 (four) times. Microscope settings were kept constant throughout. Mean intensity values obtained from HCR experiments were of 190000 for the WT embryos, with an average cell size of 85 pixels, making the average signal 50% of the maximum.

## Sharpness index

We defined a sharpness index as the ratio between the length of a perfectly sharp boundary and the actual measured length of the boundary according to the following equation:

$$S = \frac{A^{sharp}}{A^{real}} = \frac{\sum_{n=1}^{N}(d_n^{sharp} \times z)}{\sum_{n=1}^{N}(d_n^{xy} \times z)} = \frac{(\sum_{n=1}^{N} d_n^{sharp}) \times z}{(\sum_{n=1}^{N} d_n^{xy}) \times z} = \frac{\sum_{n=1}^{N} d_n^{sharp}}{\sum_{n=1}^{N} d_n^{xy}}$$

Where $S$ is the sharpness index, $A^{sharp}$ is the area of the theoretical sharp boundary, $A^{real}$ is the real measured area of the boundary, $n$ is each individual slice in the z-stack, $N$ is the total number of slices of the z-stack, $d^{sharp}$ is the minimum distance between the rhombomere's lateral edges (*Figure 4—figure supplement 1B*) in XY (the theoretical sharp boundary), $d^{xy}$ is the measured distance in *XY* of the boundary and *Z* is the thickness of each slice.

## qPCR

Eight embryos injected for HCR were separated after dechorionation and total RNA extracted with 150 µl of Trizol reagent. After chloroform addition and separation of the aqueous phase, the samples were concentrated using the DNA-Free RNA Kit (Zymo Research, Irvine, CA). Poly A-RNA was transcribed using SuperScriptII. SYBR Green qPCR reactions were performed with primers 5'-ATCTA TTCGGTGGACGAGC-3' and 5'-TAATCAGGCCATCTCCTGC-3' for Krox20 and 5'-CAAGGGA TGGAAGATTGAGC-3' and 5'-AACCATACCAGGCTTGAGGA-3' for EF1α. Primer sets were tested

**Table 1.** Sequences of the probes used for HCR corresponding to the specific genes as indicated and flanked by the corresponding adaptor sequences (B1 or B2). P1, P2, etc. corresponds to the different probes used for each gene.

| Probe | Sequence |
|---|---|
| Krox20_B1-P1 | 5'-GAGGAGGGCAGCAAACGGGAAGAGTCTTCCTTTACGATATT AGAAGTGGCTGGGGGAGACTGAGGATGCAGGTGACGAGGATGCTGAGGAT ATATAGCATTCTTTCTTGAGGAGGGCAGCAAACGGGAAGAG-3' |
| Krox20_B1-P2 | 5'-GAGGAGGGCAGCAAACGGGAAGAGTCTTCCTTTACGATATT GTGGAAAGGAACGCAGACGGGTCTTGATAGACCTCTCCGCATCCAGAGTA ATATAGCATTCTTTCTTGAGGAGGGCAGCAAACGGGAAGAG-3' |
| Krox20_B1-P3 | 5'-GAGGAGGGCAGCAAACGGGAAGAGTCTTCCTTTACGATATT AGGTTGGAAAAAGCCGGCGTAGTCCGGGATTATAGGGAACAACCCAGAGT ATATAGCATTCTTTCTTGAGGAGGGCAGCAAACGGGAAGAG-3' |
| Krox20_B1-P4 | 5'-GAGGAGGGCAGCAAACGGGAAGAGTCTTCCTTTACGATATT GTTAGAGGAGGCGGTAATTTGAAAGAGTCCAGCGGGCAGGAGAACGGTTT ATATAGCATTCTTTCTTGAGGAGGGCAGCAAACGGGAAGAG-3' |
| Hoxb1a_B2-P1 | 5'-CCTCGTAAATCCTCATCAATCATCCAGTAAACCGCCAAAAA AGTGTGGAAAGGGCCCGGGAACGCCTGGTCCAAGTGGTGGTATCCAGCCT AAAAAAGCTCAGTCCATCCTCGTAAATCCTCATCAATCATC-3' |
| Hoxb1a_B2-P2 | 5'-CCTCGTAAATCCTCATCAATCATCCAGTAAACCGCCAAAAA CAGTTCCACCATAGGTAAGGCCCATGCCAGTTTGATTTTGGTGCTGGTGA AAAAAAGCTCAGTCCATCCTCGTAAATCCTCATCAATCATC-3' |
| Hoxb1a_B2-P3 | 5'-CCTCGTAAATCCTCATCAATCATCCAGTAAACCGCCAAAAA TGTTGAGCATAGTCCGAGTTGGCGCAGGCCTGTGTCCCATAACTTGTTGT AAAAAAGCTCAGTCCATCCTCGTAAATCCTCATCAATCATC-3' |
| Hoxb1a_B2-P4 | 5'-CCTCGTAAATCCTCATCAATCATCCAGTAAACCGCCAAAAA AGTACGCACCGGCCATAGAGCCATAGTGTGGACTGGCATTTGATGTTGAA AAAAAAGCTCAGTCCATCCTCGTAAATCCTCATCAATCATC-3' |
| Hoxb1a_B2-P5 | 5'-CCTCGTAAATCCTCATCAATCATCCAGTAAACCGCCAAAAA GAGTGATCAGATTGATCCTCGAGGTCTTTAGACGAAGTGGAGGAAGCAGG AAAAAAGCTCAGTCCATCCTCGTAAATCCTCATCAATCATC-3' |

and confirmed to have an amplification efficiency of 2. In order to study the individual variability in gene expression on each embryo, a homogeneous standard was generated with RNA pooled from 100 embryos at the same stage (3 somites-11 hpf-) of development and reactions of this 'standard' were run in parallel. The $\triangle\triangle$Ct method was used to analyze the samples. All samples were run in triplicates and the experiment was repeated 4 (four) times.

## Statistical analysis

Unless otherwise noted, statistical analysis was performed using Prism 5 (GraphPad software, La Jolla, CA). In experiments where the graded distribution was analyzed (*Figure 1*, *2A*; *Figure 1—figure supplement 1C*), a comparison of the best-fit lines was performed.

In assays where the variance was analyzed (*Figure 2*, *3*, *4* and associated figure supplements) a one-way ANOVA of the coefficients of variation was performed. In order to establish significance in the changes in variation, a Levene's test was performed using MATLAB (MathWorks, Natick, MA). To establish the significance of the changes in mean values, a Newman-Keuls test was used.

For the rescue experiment (*Figure 1—figure supplement 1D–F*) a two-way ANOVA analysis was performed and significance was established after the Bonferroni post test correction.

To establish the predominant frequency of the noise (*Figure 2C*) an analysis of the datasets for different cells was performed using a moving window and searching for lags for which a correlation function would provide significant p-values using MATLAB (MathWorks, Natick, MA). Lags 13 and 14 gave p-values between 0.08 and 0.01. Average of these lags corresponds to a frequency of about 2.7 min.

Boundary sharpness was calculated as the ratio between the length of the theoretical sharp boundary and the actual measured length of the boundary (*Figure 4—figure supplement 1B*). All

perturbations yielded significant differences from wild-type controls. Thus no asterisks were included to indicate columns representing statistical significance.

## Acknowledgements

The authors thank Arthur Lander for critical reading of the manuscript and funding, Arul Subramanian and Ines Gehring for assistance with HCR, as well as Scott Fraser, Zeba Wunderlich, Catherine McCusker and members of the Schilling lab for their constructive criticisms. This work was partly supported by grants from the NIH/NIGMS to QN and TS (R01-GM107264), MD and EG (P41-GM103540) and MD, EG, QN, and TS (P50-GM76516).

## Additional information

### Funding

| Funder | Grant reference number | Author |
| --- | --- | --- |
| National Institutes of Health | P50-GM76516 | Julian Sosnik<br>Likun Zheng<br>Christopher V Rackauckas<br>Michelle Digman<br>Enrico Gratton<br>Qing Nie<br>Thomas F Schilling |
| National Institutes of Health | P41-GM103540 | Enrico Gratton<br>Michelle Digman |
| National Institutes of Health | R01-GM107264 | Qing Nie<br>Thomas F Schilling |

The funders had no role in study design, data collection and interpretation, or the decision to submit the work for publication.

### Author contributions

JS, EG, QN, Conception and design, Acquisition of data, Analysis and interpretation of data, Drafting or revising the article; LZ, Conception and design, Analysis and interpretation of data; CVR, Analysis and interpretation of data, Drafting or revising the article; MD, Acquisition of data, Analysis and interpretation of data; TFS, Conception and design, Analysis and interpretation of data, Drafting or revising the article

### Author ORCIDs

Christopher V Rackauckas, http://orcid.org/0000-0001-5850-0663
Thomas F Schilling, http://orcid.org/0000-0003-1798-8695

### Ethics

Animal experimentation: The study was performed in strict accordance to the recommendations in the Guide for the Care and Use of Laboratory Animals of the National Institutes of Health. All of the animals were handled according to approved institutional animal care and use committee (IACUC) protocols (#2000-2149) of the University of California, Irvine. The renewal of this protocol was approved by the IACUC (Animal Welfare Assurance #A3416.01) on December 11, 2015. All animal experiments were performed on embryos derived from natural breedings and very effort was made to minimize suffering.

## Additional files

### Supplementary files

• Supplementary file 1. (A) Stochastic mathematical modeling. (B) Modeling parameters

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
