## [Decision Letter]

[Editors’ note: a previous version of this study was rejected after peer review, but the authors submitted for reconsideration. The first decision letter after peer review is shown below.]

Thank you for submitting your work entitled "Noise modulation in retinoic acid signaling sharpens segmental boundaries of gene expression in the zebrafish hindbrain" for consideration by *eLife*. Your article has been reviewed by three peer reviewers, and the evaluation has been overseen by a Reviewing Editor and Janet Rossant as the Senior Editor.

The reviewers have discussed the reviews with one another and the Reviewing and Senior Editor have drafted this decision.

This is an interesting manuscript that uses FLIM microscopy for the direct visualization of retinoid concentration gradients in the zebrafish hindbrain. These finding provide insight into the dynamic spatial heterogeneity of RA and its implications for hindbrain patterning. Despite the interesting set of experiments, the paper had several weaknesses in presentation and analysis of data, which we anticipate will require extensive reanalysis, and the generation of additional data. Since it is *eLife* policy to only invite revisions in cases where revisions can be reasonably completed within two months, we cannot invite revision of the current manuscript. However, we are in principle interested in the study, and we would consider a substantially revised version that addresses our concerns, with the proviso that it will be treated as a new submission.

The key revisions that came from the discussions among the reviewers are:

1) Indicate if their RA measurements show significant shifts in the position of the anterior boundary of RA as time proceeds (as was seen with RARE-lacZ in mouse where RA activity is first at the r2/r3 boundary, then at r3/r4, then at r4/r5) or do they see a more noisy decline in RA as time proceeds since they are detecting RA itself rather than RA activity. (This may require more imaging of the gradients).

2) Describe the shape of the RA relative abundance in the zebrafish hindbrain as a function of A-P position.

3) Show single cell resolution at the presumptive rhombomere boundaries. They also need quantify the HCR data in a linear range and at the level of single cells. (Perform more HCR in situ experiments).

4) How do the authors determine S without the *hoxb1* expression or another marker that defines the adjacent rhombomere segment? The sharpness index (S) in their Zhang Mol Sys Biol 2012 paper defined as 'the mean location of the boundary between *hoxb1* and *krox20* expression domains as the intersection of their distributions at 50% of the normalized value. A decrease in S over time indicates noise attenuation and a sharper boundary. (Perform more imaging and in situ experiments).

Reviewer #1:

This manuscript provides important new information on the mechanism of retinoic acid (RA) signaling during hindbrain development. Determination of RA concentration gradients has been a difficult task, hampering efforts to understand RA function. Here, the authors present a novel methodology utilizing fluorescence lifetime imaging microscopy to determine the abundance of endogenous RA along the A-P axis of the zebrafish hindbrain. They show that inherent noise exists in the concentration of RA in individual cells of the hindbrain, and that CRABP2 (an RA-binding protein) attenuates the noise to allow proper expression of downstream targets. Overall, these studies have significantly increased our understanding of RA gradient action during hindbrain formation.

Specific Points to Address:

1) Paragraph one, subheading “Fluorescence Lifetime Imaging (FLIM) measures endogenous RA directly in vivo” – the authors could add that their observation of a declining RA concentration from posterior to anterior hindbrain is consistent with the pattern of RARE-lacZ expression along the hindbrain of late-gastrulation stage mouse embryos (Sirbu, Development 2005).

2) Paragraph two, subheading “Fluorescence Lifetime Imaging (FLIM) measures endogenous RA directly in vivo” – how do the authors results on differential RA concentration along the hindbrain A-P axis over time compare with results from mouse RARE-lacZ suggesting that RA activity experiences shifting hindbrain boundaries from anterior to posterior as development proceeds (Sirbu Development 2005)?

3) Paragraph two, subheading “Functional roles for noise in RA target gene expression” – it would help if the authors further discuss whether their analysis of Cyp26a1 (an RA-degrading enzyme) supports a role for Cyp26 in generating rhombomere boundaries like suggested in other studies (such as Hernandez, Development 2007).

*Reviewer #2:*in vivo visualization of retinoic acid (RA) by light microscopy has proved challenging due to the non-peptidic structure of RA, its low abundance in cells, and wide spectra of absorbance and emission. This has made it difficult to address the specific question of whether an RA gradient exists during segmentation of the vertebrate hindbrain into distinct rhombomeres, and if so how noisy RA levels in the hindbrain are modulated by cells in order to refine gene expression at rhombomere segment boundaries.

In this paper, the authors present an excellent application of fluorescence lifetime microscopy (FLIM) and phasor analysis to more accurately measure endogenous RA directly in the living zebrafish embryo, examine noise in RA signaling, and investigate how modulation of this noise (by intracellular RA-binding proteins (Crabps) or RA-degrading enzymes (Cyp26s)) affects patterning of hindbrain segments using fluorescence in-situ hybridization chain reaction technology (HCR) to more clearly define rhombomere-specific gene expression (Krox20). The authors clearly show their Phasor-FLIM strategy is accurate in vivo.

However, there are several significant concerns regarding the data presented that limit my enthusiasm for their primary conclusions and publication.

1) The authors argue this is the first method to directly measure RA concentration levels in the living embryo and only briefly mention a previous FRET-reporter based approach which makes this claim (Shimozono et al., Nature, 2013) and was used to measure RA concentration in the zebrafish hindbrain. This can easily be fixed with more description in the Introduction and a revision of the wording. However, what is troubling here is that the authors appear to move back and forth between two distinctly different tissue architectures (neural ectoderm (2D) vs hindbrain (3D) in their measurements of RA concentration and noise levels. That is, we don't know the shape of the RA relative abundance in the zebrafish hindbrain as a function of A-P position. Also, the result that Crabps modulates noise in RA is again based on measurements made in the neural ectoderm and not the hindbrain.

2) HCR is a more sensitive method to detect mRNA expression that the authors use to examine sharpening/fuzziness of Krox20 expression in response to changes in RA levels. This is precisely the type of experiment to perform in response to their computational modeling simulations of noise-induced switching and boundary sharpening. Unfortunately, the authors completely under-utilize this method and this brings their data and conclusion into question. That is, the HCR images (z-projections) are over-exposed and should be of high enough quality to present an accurate 3D spatial representation of the subregions near presumptive rhombomere boundaries (instead of reducing to a 2D analysis and segmentation that does not include the cell membrane outlines as a 3D volume). Further, this conclusion really begs for single cell resolution at the presumptive rhombomere boundaries which is not clear. Overall, this is a poor presentation compared with the more elegant Phasor-Flim analysis.

Reviewer #3:

The manuscript by Sosnik et al. uses fluorescence lifetime imaging microscopy (FLIM) and phasor analysis to measure endogenous retinoic acid (RA) levels within zebrafish embryos. The authors studied the amplitude of noise in RA signaling, and how modulation of this noise affects rhombomere patterning. The authors argue that RA forms a noisy gradient during critical stages of hindbrain patterning and that cells use distinct intracellular binding proteins to attenuate noise in RA levels. Increasing noise disrupts sharpening of rhombomere boundaries and proper patterning of the hindbrain. The work is very interesting and innovative. The general concerns are that the experiments are difficult to understand and it appears important controls are lacking. We would support the paper being accepted for publication if these issues are addressed.

In the subsection “Functional roles for noise in RA target gene expression”, the authors state: “To determine how altering RA levels influences noise in gene expression within the hindbrain we disrupted Crabp2a or Cyp26a1 and assayed expression of *krox20* in r3 and r5.”

Each MO treatment seems to have had an effect on *krox20* expression. Considering that MO could have broad effects on RNA expression levels, where any MOs unrelated to the RA pathway injected and analyzed.

It is not apparent that HCR is a reliable indicator of gene expression levels. This is a novel USE of HCR, so proper controls should be considered. What other RNAs besides *krox20* were measured? Were other RNAs that should not be affected and presumed to be affected by altering RA levels within the hindbrain also examined? These could also be measured within the other rhombomeres.

Is it possible to use the Tg(shhb:KalTA4,UAS-E1b:mCherry) transgenic zebrafish to compare cherry protein vs. RNA expression levels in the rhombomeres as a quick control?

The HCR images shown in Figure 4 appear saturated, which would prevent accurate quantitation of *krox20* RNA levels.

[Editors’ note: what now follows is the decision letter after the authors submitted for further consideration.]

Thank you for resubmitting your work entitled "Noise modulation in retinoic acid signaling sharpens segmental boundaries of gene expression in the zebrafish hindbrain" for further consideration at *eLife*. Your revised article has been favorably evaluated by Janet Rossant (Senior editor), a Reviewing editor, and three reviewers. The manuscript has been improved but there are some remaining issues that need to be addressed before acceptance, as outlined below:

The reviewers have read and discussed your manuscript and in principle feel it is worthy of publication if you can made a few minor changes and address one major area of concern that needs to be clarified.

Major Concern:

The major concern deals centers about the HCR analyses. It is the opinion of the reviewers that the authors’ claim of using HCR for the relative quantification of target mRNAs is novel and important and thus requires solid data to prove that their approach is indeed quantitative. As presented the HCR experiments are difficult to understand and important controls are lacking. The authors should include 1 control data point. That is, no initiator probe + hairpins to show that there is no non-specific signal.

The authors responded to the reviewers’ previous comments with the following statement:

“HCR is an established technique that linearly amplifies the signal of an RNA target (Dirks and Pierce, 2004; molecularinstruments.org; Choi et al., 2010; 2014). As mentioned above, we also analyzed the RNA of *hoxb1*a in r3, r4 and r5 (only r3 and r5 data are presented here – Figure 6) at early stages when its expression is low and uniform across the hindbrain field. The MO and RNA injections had no significant effects on *hoxb1*a expression as assayed by HCR.”

There is disagreement with this response. Choi et al. 2014 (ACS Nano 8, 4284) is the most relevant paper since it describes the use of DNA probes to detect RNA targets and describes the method followed by the authors. However, the Choi paper does not demonstrate HCR's capability of doing relative (or absolute) quantification of target mRNAs. Choi et al. compared signal intensities and signal-to-background between one- and two-initiator probes using multiplex approaches. Thus, the authors’ claim of using HCR for the relative quantification of target mRNAs is novel and important and thus requires a higher burden of proof that it is indeed quantitative.

The basic experimental framework put forth by the authors is based upon the idea of imaging the HCR signals from the tissue/embryonic region of interest (hindbrain r3-5) and examining whether HCR provides a quantitative readout of relative mRNA abundance within single cells for a statistically significant number of image sets. Determining relative intensities using HCR can be complicated by several factors that contribute to background signals (ACS Nano 8, 4284). Auto-fluorescence (AF) in the embryos may prevent an experimenter from observing the true intensities directly in the experiments. Non-specific amplification (NSA) of hairpins and non-specific detection (NSD) of the targets can also corrupt the true signal measurement. All of these controls were presumably run by the authors and would be valuable additions to the paper.

---

## [Author Response]

[Editors’ note: the author responses to the first round of peer review follow.]

Thank you for considering our manuscript for review entitled “Noise modulation in retinoic acid signaling sharpens segmental boundaries of gene expression in the zebrafish hindbrain”. We have addressed all of the comments, both experimentally and in textual revisions. We appreciate the importance of the more detailed spatial and temporal analyses requested both by reviewers and editors (Editor’s comments #1) and we have addressed these experimentally and/or computationally where possible. However, as we explain below there are no morphological landmarks of individual rhombomeres at the stages when most of our measurements were performed, and fluorescent transgenes that mark rhombomeres at late gastrula stages (when segments are sharpening) interfere with the RA emission and therefore with the FLIM measurements. The FLIM technique is based on spontaneous autofluorescence of RA and any fluorescent marker will strongly perturb the basic FLIM- phasor fluctuation analysis. We have clarified these limitations of the technique in the text, and we have managed to perform a more detailed analysis of gradient shape with additional FLIM and computational analyses, as requested (Editors comment #2). We also now include data to address concerns regarding quantification of the HCR data in 3D and at the level of single cells (Editors comment #3), as well as the lack of controls for these experiments (Reviewer 3, major comment #1). Overall, the reviewers agree that these are novel findings that improve our understanding about the function of morphogen gradients and an understudied dimension of cell signaling. Following are our specific responses to the points raised by editors and reviewers to our previously submitted manuscript.

*The key revisions that came from the discussions among the reviewers are:*

*1) Indicate if their RA measurements show significant shifts in the position of the anterior boundary of RA as time proceeds (as was seen with RARE-lacZ in mouse where RA activity is first at the r2/r3 boundary, then at r3/r4, then at r4/r5) or do they see a more noisy decline in RA as time proceeds since they are detecting RA itself rather than RA activity. (This may require more imaging of the gradients).*

As we discussed in the overall response, we would love to do these more detailed spatial and temporal analyses. However, fluorescent transgenes that mark rhombomeres at these early stages interfere with FLIM measurements of spontaneous RA autofluorescence. We tried to address the issue of posterior shifts in the gradient by collecting a series of successive FLIM measurements within individual live embryos, leaving them embedded in agar on the microscope to retain the field of view, but this failed because of the excessive laser exposure. These technical limitations were not clearly discussed in the original manuscript and are now clarified in the text (paragraph one, subheading “Fluorescence Lifetime Imaging (FLIM) measures endogenous RA directly in vivo”).

*2) Describe the shape of the RA relative abundance in the zebrafish hindbrain as a function of A-P position.*

This is an excellent point. We evaluated gradient shape in more detail with additional FLIM measurements and a least squares regression analyses (new Figure 1—figure supplement 2). Exponential, linear and quadratic curve fits were produced for the entire field, but because each of these fits explains similar amounts of the variance it is hard to differentiate between the curve types. We also tried to see if the shape of the gradient differs between anterior and posterior, by splitting the FLIM dataset at the 310 um point along the A-P axis and computed the fits on the two portions. The results were almost identical to the full fit and are shown in Figure 1—figure supplement 2, panels B and C. While it is difficult to distinguish between exponential, linear, and quadratic functions, the measured RA clearly fits a shape with characteristics shared by all three, strongly confirming the presence of a spatial gradient of RA. This is now discussed in the text (paragraph one, subheading “Fluorescence Lifetime Imaging (FLIM) measures endogenous RA directly in vivo”)

*3) Show single cell resolution at the presumptive rhombomere boundaries. They also need quantify the HCR data in a linear range and at the level of single cells. (Perform more HCR in situ experiments).*

All HCR analyses were performed on raw 3D data and we can provide movies showing the 3D volumes at the editors/reviewers’ discretion. Z-projections and contrast enhancement were performed post-analysis to simplify presentation. This has been clarified in the revised manuscript (paragraph one, subheading “Functional roles for noise in RA target gene expression”; see also response to reviewer #2, item 2). At the onset of *krox20* expression, levels are very low. Raw projections are presented here in Figure 5, which we can add as a supplement or replace Figure 4 (top panel) at the editors/reviewers’ discretion. This has also been further clarified in the figure legend.

*4) How do the authors determine S without the hoxb1 expression or another marker that defines the adjacent rhombomere segment? The sharpness index (S) in their Zhang Mol Sys Biol 2012 paper defined as 'the mean location of the boundary between hoxb1 and krox20 expression domains as the intersection of their distributions at 50% of the normalized value. A decrease in S over time indicates noise attenuation and a sharper boundary. (Perform more imaging and in situ experiments).*

In Zhang et al. 2012, (Figure 4) we compared the sharpness index (S) calculated based on models incorporating both *krox20* and *hoxb1*a with the sharpness measured experimentally using *krox20* expression alone and found them to be in good agreement. Therefore, in this manuscript we measure the boundary between *krox20* positive and negative cells. This is now better explained in the text (see response to Reviewer #2, item 3).

Reviewer #1:

*Specific Points to Address:*

*1) Paragraph one, subheading “Fluorescence Lifetime Imaging (FLIM) measures endogenous RA directly in vivo” – the authors could add that their observation of a declining RA concentration from posterior to anterior hindbrain is consistent with the pattern of RARE-lacZ expression along the hindbrain of late-gastrulation stage mouse embryos (Sirbu, Development 2005).*

We have added the Sirbu et al. (2005) reference (paragraph two, Introduction).

*2) Paragraph two, subheading “Fluorescence Lifetime Imaging (FLIM) measures endogenous RA directly in vivo” – how do the authors results on differential RA concentration along the hindbrain A-P axis over time compare with results from mouse RARE-lacZ suggesting that RA activity experiences shifting hindbrain boundaries from anterior to posterior as development proceeds (Sirbu Development 2005)?*

We address this in the Response to Editor’s comments #1.

*3) Paragraph two, subheading “Functional roles for noise in RA target gene expression”* – *it would help if the authors further discuss whether their analysis of Cyp26a1 (an RA-degrading enzyme) supports a role for Cyp26 in generating rhombomere boundaries like suggested in other studies (such as Hernandez, Development 2007).*

Additional discussion of Cyp26s has been added, as well as the Hernandez et al. (2007) reference (Introduction).

Reviewer #2:

*1) The authors argue this is the first method to directly measure RA concentration levels in the living embryo and only briefly mention a previous FRET-reporter based approach which makes this claim (Shimozono et al., Nature, 2013) and was used to measure RA concentration in the zebrafish hindbrain. This can easily be fixed with more description in the Introduction and a revision of the wording.*

While the FRET-reporters for RA (GEPRAs) directly bind RA, the measurements made in that paper are of the fluorescent FRET reporters and therefore inherently indirect. Our methodology directly measures the endogenous autofluorescence of RA, making it the first truly direct method. The text has been modified (subheading “FLIM measurements reveal high noise levels in RA”).

*However, what is troubling here is that the authors appear to move back and forth between two distinctly different tissue architectures (neural ectoderm (2D) vs hindbrain (3D) in their measurements of RA concentration and noise levels. That is, we don't know the shape of the RA relative abundance in the zebrafish hindbrain as a function of A-P position. Also, the result that Crabps modulates noise in RA is again based on measurements made in the neural ectoderm and not the hindbrain.*

The “neural ectoderm” in this case refers to the region of the ectoderm of the gastrula stage embryo that will form the hindbrain, based on previous fate maps generated at mid-gastrula stages (8-8.5 hpf; Woo and Fraser (1995) – Development 121, 2595). We (and others) have shown previously that these are the critical stages for RA in hindbrain segmentation (White et al., 2007; Hernandez et al., 2007). FLIM measurements are 2D and performed on individual confocal slices. FLIM requires a high numerical aperture objective (40X/NA 1.2) – that intrinsically has a short working distance – making it impossible to perform the measurements at lower magnification. Fortunately at these stages the neural ectoderm is extremely thin (2-3 cells thick). An explanation of the limitations of the technique has been added to the Phasor-FLIM supplementary text (paragraph one, subheading “Phasor-FLIM”).

*2) HCR is a more sensitive method to detect mRNA expression that the authors use to examine sharpening/fuzziness of Krox20 expression in response to changes in RA levels. This is precisely the type of experiment to perform in response to their computational modeling simulations of noise-induced switching and boundary sharpening. Unfortunately, the authors completely under-utilize this method and this brings their data and conclusion into question. That is, the HCR images (z-projections) are over-exposed and should be of high enough quality to present an accurate 3D spatial representation of the subregions near presumptive rhombomere boundaries (instead of reducing to a 2D analysis and segmentation that does not include the cell membrane outlines as a 3D volume). Further, this conclusion really begs for single cell resolution at the presumptive rhombomere boundaries which is not clear. Overall, this is a poor presentation compared with the more elegant Phasor-Flim analysis.*

We performed all HCR analyses on raw 3D data. Movies showing the 3D volume of representative datasets can be included at the editors/reviewers’ discretion. We generated Z-projections and enhanced contrast post-analysis to simplify presentation. At the onset of *krox20* expression, levels are very low and the background is high. Raw projections are presented here in Figure 5, which we can add as a supplement or replace Figure 4 (top panel) at the editors/reviewers’ discretion. This has also been further clarified in the figure legend.

Author response image 1.Raw (not contrast enhanced) z-projections of the 3D stacks utilized to quantify the HCR data (Krox20 gene expression and boundary sharpness).**DOI:**
http://dx.doi.org/10.7554/eLife.14034.015

Reviewer #3:

*In the subsection “Functional roles for noise in RA target gene expression”, the authors state: “To determine how altering RA levels influences noise in gene expression within the hindbrain we disrupted Crabp2a or Cyp26a1 and assayed expression of krox20 in r3 and r5.”*

*Each MO treatment seems to have had an effect on krox20 expression. Considering that MO could have broad effects on RNA expression levels, where any MOs unrelated to the RA pathway injected and analyzed.*

Due to the high cost of MO and HCR reagents, we did not perform experiments with MOs unrelated to the RA pathway. However, the effects of the Crabpa2 and Cyp26a1 had distinct effects on RA noise levels that were opposite to the overexpression obtained by RNA injections. Furthermore, we analyzed the effects of these MOs and RNAs on another RA target gene (*hoxb1*a) by HCR and found that they caused no significant changes (Figure 6). These results can be included as a part of Figure 4 —figure supplement if the editors/reviewers prefer.

Author response image 2.Mean-centered analysis of *hoxb1*a expression of a subset of cells for r3 and r5 from 3 randomly selected embryos for each indicated treatment.The results show no significant differences between the treatments and the WT controls**DOI:**
http://dx.doi.org/10.7554/eLife.14034.016

*It is not apparent that HCR is a reliable indicator of gene expression levels. This is a novel USE of HCR, so proper controls should be considered. What other RNAs besides krox20 were measured? Were other RNAs that should not be affected and presumed to be affected by altering RA levels within the hindbrain also examined? These could also be measured within the other rhombomeres.*

HCR is an established technique that linearly amplifies the signal of an RNA target (Dirks and Pierce, 2004; molecularinstruments.org; Choi et al., 2010; 2014). As mentioned above, we also analyzed the RNA of *hoxb1*a in r3, r4 and r5 (only r3 and r5 data are presented here – Figure 6) at early stages when its expression is low and uniform across the hindbrain field. The MO and RNA injections had no significant effects on *hoxb1*a expression as assayed by HCR.

*Is it possible to use the Tg(shhb:KalTA4,UAS-E1b:mCherry) transgenic zebrafish to compare cherry protein vs. RNA expression levels in the rhombomeres as a quick control?*

As mentioned above, we include *hoxb1*a as the control to address this concern.

*The HCR images shown in Figure 4 appear saturated, which would prevent accurate quantitation of krox20 RNA levels.*

We agree and address this as described in the response to Reviewer 2, comment #2 and in Figure 5.

[Editors' note: the author responses to the re-review follow.]

*Major Concern:*

*The major concern deals centers about the HCR analyses. It is the opinion of the reviewers that the authors’ claim of using HCR for the relative quantification of target mRNAs is novel and important and thus requires solid data to prove that their approach is indeed quantitative. As presented the HCR experiments are difficult to understand and important controls are lacking. The authors should include 1 control data point. That is, no initiator probe + hairpins to show that there is no non-specific signal.*

We appreciate the concern raised by the reviewers and we have done this control, which produces no non-specific signal (at precisely the same settings on the Leica confocal microscope that was used for all of our images), which we now mention on subheading “Hybridization Chain Reaction (HCR)”. We have also expanded the description of HCR in the Materials and methods and include a 3D rendering (Video 1). Further evidence for specificity in the hairpins includes: 1) the signal outside of rhombomeres 3 and 5 (for Krox20 probes) is very low and evenly distributed along the sample (see Video 1), 2) single initiator controls give comparable background, and 3) Hoxb1a initiators show a distinct pattern.

*The authors responded to the reviewers’ previous comments with the following statement:*

*“HCR is an established technique that linearly amplifies the signal of an RNA target (Dirks and Pierce, 2004; molecularinstruments.org; Choi et al., 2010; 2014). As mentioned above, we also analyzed the RNA of hoxb1a in r3, r4 and r5 (only r3 and r5 data are presented here – Figure 6) at early stages when its expression is low and uniform across the hindbrain field. The MO and RNA injections had no significant effects on hoxb1a expression as assayed by HCR.” There is disagreement with this response. Choi et al. 2014 (ACS Nano 8, 4284) is the most relevant paper since it describes the use of DNA probes to detect RNA targets and describes the method followed by the authors. However, the Choi paper does not demonstrate HCR's capability of doing relative (or absolute) quantification of target mRNAs. Choi et al. compared signal intensities and signal-to-background between one- and two-initiator probes using multiplex approaches. Thus, the authors’ claim of using HCR for the relative quantification of target mRNAs is novel and important and thus requires a higher burden of proof that it is indeed quantitative. The basic experimental framework put forth by the authors is based upon the idea of imaging the HCR signals from the tissue/embryonic region of interest (hindbrain r3-5) and examining whether HCR provides a quantitative readout of relative mRNA abundance within single cells for a statistically significant number of image sets. Determining relative intensities using HCR can be complicated by several factors that contribute to background signals (ACS Nano 8, 4284). Auto-fluorescence (AF) in the embryos may prevent an experimenter from observing the true intensities directly in the experiments. Non-specific amplification (NSA) of hairpins and non-specific detection (NSD) of the targets can also corrupt the true signal measurement. All of these controls were presumably run by the authors and would be valuable additions to the paper.*

These issues have now been addressed in the expanded HCR description in Materials and methods.